# Polyphenolic Profile of Tunisian Thyme (*Thymbra capitata* L.) Post-Distilled Residues: Evaluation of Total Phenolic Content and Phenolic Compounds and Their Contribution to Antioxidant Activity

**DOI:** 10.3390/molecules27248791

**Published:** 2022-12-12

**Authors:** Kheiria Hcini, Abir Bahi, Monia Bendhifi Zarroug, Mouna Ben Farhat, Antonio Abel Lozano-Pérez, José Luis Cenis, María Quílez, Sondes Stambouli-Essassi, Maria José Jordán

**Affiliations:** 1Biodiversity, Biotechnology and Climate Change Laboratory (LR11ES09), Department of Life Sciences, Faculty of Science of Tunis, University of Tunis El Manar, Tunis 2092, Tunisia; 2Department of Life Sciences, Faculty of Sciences of Gafsa, University Campus Sidi Ahmed Zarroug, University of Gafsa, Gafsa 2112, Tunisia; 3Laboratory of Analysis, Treatment and Valorisation of Pollutants of the Environment and Products, Faculty of Pharmacy, University of Monastir, Monastir 5000, Tunisia; 4Department of Biological Engineering, Research Unit of Active Biomolecules Valorisation, Higher Institute of Applied Biology of Medenine , University of Gabes, Gabes 6029, Tunisia; 5Laboratoire des PlantesExtrêmophiles, Centre de Biotechnologie de Borj-Cédria, BP 901, Hammam-Lif 2050, Tunisia; 6Departamento de Biotecnología, Genómica y Mejora Vegetal, Instituto Murciano de Investigación y DesarrolloAgrario y Medioambiental (IMIDA), La Alberca, 30150 Murcia, Spain; 7Instituto Murciano de InvestigaciónBiosanitaria (IMIB)-Arrixaca, El Palmar, 30120 Murcia, Spain; 8Departamento de Desarrollo Rural, Enología y Agricultura Sostenible, Instituto Murciano de Investigación y Desarrollo Agrario y Medioambiental (IMIDA), La Alberca, 30150 Murcia, Spain

**Keywords:** *Thymbra capitata* L., post-distilled residues, total polyphenolic content, antioxidant activity, DPPH, FRAP, polyphenolic compounds, HPLC, correlation

## Abstract

During the last decade there has been growing interest in the formulation of new cosmetic, food and pharmaceutical products containing natural compounds with antioxidant activity and other beneficial properties. Aromatic and medicinal plants have always been the major source of bioactive compounds, especially, wild thyme (*Thymbra capitata* L.), which has been used since ancient times for its valuable health benefits that could be attributed to the richness of polyphenolic compounds. This study was undertaken with the following aims: to estimate the total polyphenolic content (TPC); to evaluate the antioxidant activity; to identify and quantify the phenolic compounds of post-distilled residues of Tunisian thyme, and their contribution to the antioxidant activity. The TPC, as determined by the Folin–Ciocalteu method, was found to reach the values of 126.7 and 107.84 mg gallic acid equivalent/g plant dry weight (mg GAE/g PDW). The antioxidant activity, which is assessed by DPPH and FRAP assays, reached the values of 42.97–45.64 μg/mL and 42.22–50.21 mMFe^2+^/mg PDW, respectively. HPLC analysis revealed the presence of fourteen polyphenolic compounds, of which diosmin and rosmarinic acid were found to be the most abundant (24.26 to 33.80 and 22.0.1 to 26.29 mg/g PDW, respectively). An important correlation was found between the antioxidant activity and several identified phenolic compounds (*p* < 0.05). The findings revealed that thyme post-distilled residues have an effective natural antioxidant potential due to their high concentration of bioactive molecules, and they appear to be useful in the pharmaceutical, cosmetic, and food industries, with beneficial effects on human health. Therefore, supplementing a balanced diet with herbs may have beneficial health effects.

## 1. Introduction

In recent decades, there has been great interest in the use of herbs and plants, as they have been the sources of natural products, commonly named as bioactive compounds, with several beneficial activities [1]. The biological, pharmacological and medicinal properties of this group of compounds have been extensively reviewed, mainly for their antioxidant properties [2]. Infact, several studies have indicated that the consumption of natural antioxidant compounds protect cells against the damage of reactive oxygen species such as singlet oxygen, superoxide, peroxyl radicals, hydroxyl radicals and peroxynitrite [3]. Numerous studies have demonstrated that some herbs and plant extracts are as efficient as synthetic antioxidants such as butylated hydroxytoluene (BHT) and butylated hydroxyanisole (BHA), which have carcinogenic effects on living organisms [4,5].

In the search of plants as a source of natural antioxidants, some aromatic and medicinal plants have been extensively studied for their richness in natural products, which have beneficial properties, namely, polyphenols, vitamins, polysaccharides and minerals [1,6]. Specifically, the natural compounds from the Lamiaceae family (thyme, sage and rosemary) have been reported in several studies for its antioxidant, anti-inflammatory, antimicrobial and anticarcinogenic activities [5,7,8,9,10]. In particular, thyme extracts possess very useful antioxidant properties, which appear to be related to their content of essential oil and phenolic compounds [11,12,13].

Wild thyme is widely used as a food item and is important for the food and pharmaceutical industries because of its health benefits related to its antioxidant, antimicrobial, antispasmodic and antihypertensive effects; it is also beneficial in treating respiratory and gastrointestinal tract disorders [9,14,15]. The genus Thymus, which belongs to the Lamiaceae family, includes 350 species widespread all around the world [12]. In Tunisian flora, this genus is mainly represented by *Thymbra capitata* L., a perennial, herbaceous shrub commonly used as a spicy herb and locally known under the common name “zaâtar” [16]. *T. capitata* L. is widely used in folk medicine as a stomachic, diaphoretic or antispasmodic, specifically for whooping cough, as a stimulant for blood circulation and as an aphrodisiac [17,18].

*T. capitata* L. is among the most promising sources used for the recovery of essential oils through hydrodistillation. Yet, the exploitation for polyphenols recovery from the residues that remain post-distillation, which may be used as antioxidants in foods, food supplements or cosmetics, is really limited. Nevertheless, some previous studies have shown that the post-distillation waste materials from thyme still possess antioxidant and antimicrobial activities [11,19,20]. Furthermore, the residues remaining after essential oil recovery, which is currently disposed of as waste, have been studied for their content of a diversity of biologically active compounds, including antioxidants such as phenolic acids and flavonoids, which is useful in increasing the shelf life of food [21,22,23,24,25]. 

Polyphenols can be found in free form or bound to plant tissues; the experimental methodology most frequently used for the quantification of total polyphenols is based on the extraction of a portion of free polyphenols, which may justify the lack of correlation often found between them and antioxidant activity [26]. The bound polyphenols are associated with polysaccharides (dietary fiber), proteins or are simply bound to each other, forming high-molecular-weight compounds, which makes their extraction difficult [27]. Therefore, it is important to develop methods that allow for the extraction of all phenolic compounds. It seems that the increase in temperature, in the range of 50–70 °C, produces a higher solubility of the compounds and increases their rate of transfer [28,29].

To the best of our knowledge, there is no previous work or information available on the chemical profile and antioxidant activity of post-distillation residues of Tunisian *T. capitata* L. aerial parts. In light of the above, the aims of this study are to determine the total polyphenolic content, to evaluate the antioxidant activity and to identify and quantify the polyphenolic compounds of post-distilled-residue extracts of thyme. Moreover, their contribution to the antioxidant activity will be investigated. This phytochemical characterization was carried out to re-valorize this wild plant by recovering polyphenols from their by-product as a source of bioactive molecules.

## 2. Results and Discussion

### 2.1. Total Polyphenolic Content

The total polyphenolic content (TPC) in the post-distillation thyme extracts reached values of 126.7± 34.3 and 107.84 ± 14.6 mg gallic acid equivalent/g plant dry weight (mg GAE/g PDW) (Table 1). These values prove that thyme residues remaining after post-distillation (hydrodistillation wastes) are rich in phenolic compounds. Our results showed a similar TPC to that obtained by Jordan et al. [11] for *Thymus zygis* subsp. *gracilis* distillation by-products. However, a recent investigation including several thyme species revealed lower amounts of total phenolics of non-distilled plant material [12,14].

Results reported by Parejo et al. [21] showed that plant material submitted to hydrodistillation is found to contain a higher number of phenolic substances than the non-distilled plant material. In certain cases, cell wall phenolics or bound phenolics can be released consequently to heat exposure, thus generating more phenolics to be extracted. These findings are consistent with data from Alu’datt et al. (2017) [30], who showed that most phenolics are found in the free form in different fruits of the Rutaceae family (85–93% of the total phenolic content).Many studies described several biological activities that aromatic plants by-products have and confirmed that these properties are directly related with the concentration of the principal components present in these polyphenolic extracts [7,11,31,32]. Accordingly, the value of the thyme residue extract as a matrix rich in phenolic substances prompts us to assess the individual polyphenolics which are abundant in its composition.

### 2.2. Antioxidant Activity

The antioxidant capacity of thyme residue extract was studied with two assays: free radical scavenging (DPPH) and reducing power (FRAP). The DPPH free radical scavenging process yielded values of 42.97 ± 2.10 and 45.64 ± 2.29 of micrograms of dry plant methanol extract per milliliter of methanol (μg/mL).The results of the FRAP test resulted in the values of 50.21 and 42.22 mMFe^2+^/mg PDW (Table 1). All plants have important antioxidant activity. These results show that the plants with high antioxidant capacity are characterized by high levels of total polyphenolic content. In this case, the antiradical activity is due to the quality of the extract, not to the quantity. Wojdyło et al. (2007) [33] reported a significant positive correlation between the antiradical activity of Lamiaceae and the total polyphenols, which demonstrates the importance of these antioxidant compounds in spices and their significant contribution to the total antioxidant activity. In addition, the search for natural antioxidants in wastes of plant origin is also being explored as an alternative to the synthetic antioxidants used in the food and pharmaceutical industries. Alternatively, by-products from essential oil production, i.e., distillate residues of aromatic plants, are a potential pool of compounds with strong antioxidant activity and can contribute to protective effects on human health. Thus, the residues remaining post-distillation of aromatic plant oils are considered as a natural source of antioxidants [7,21,25,34,35].

### 2.3. Polyphenolic Profile

The HPLC chromatogram of thyme post-distilled residues extract is shown in Figure 1. The qualitative and quantitative analysisrevealed the presence of fourteen polyphenolic compounds (Table 2). Among the mentioned phenolic compounds, diosmin and rosmarinic acid were found to be the most abundant compounds (24.26 to 33.50 and 22.01 to 26.29 mg/g PDW respectively) followed by Luteolin-7-O-Neohesperidoside, Apegenin-7-Glucoronide, Quercetin, and Olivetol. Structures of these main compounds are presented in Figure 2a,b. Previous studies have found that phenolic compounds such as rosmarinic, caffeic, ferulic, and carnosic acids, and flavonoids such as quercetin, apigenin and luteolin, were identified by Jordán et al. [11] in some species of the genus Thymus. Zheng and Wang (2001) [1] reported the analysis of phenolic components in fresh leaves of *Thymus vulgaris* L. and found that the main identified phenolic components were rosmarinic acid, caffeic acid, luteolin and hispidulin.

Published literature data confirmed the influence of the distillation process on polyphenolic contents in several Lamiaceaespecies of *Salvia officinalis* L. [36], *Rosmarinus officinalis* L. [37] and Thymus zygissubsp. Gracilis (Boiss) [11]. Almela et al. [37] reported that the drying and/or distillation treatments of plant material strongly affected the content of rosmarinic acid and carnosic acid (two compounds of strong antioxidant activity).

Considering our results, the highest values of free radical scavenging activity of thyme post-distilled residues could be due to their higher contents of phenolic components.It was the greater presence of these components that was responsible for the increased antioxidant capacity of the corresponding extracts. Consequently, it can be concluded that the concentration of the chemical compounds of polyphenolic extracts has an indispensable role in their antioxidative power. This is the first report showing the presence of an array of phenolic compounds in Tunisian wild thyme (*Thymbra capitata* L.) post-distilled residues.

### 2.4. Correlation Analysis

Table 3 shows the Pearson correlation coefficients (r) between the TPC, the identified polyphenolic compounds and the antioxidant activity, in an attempt to estimate the contribution of these compounds to the total antioxidant activity. Caffeic acid (r =0.93), rosmarinic acid (r = 0.83), luteolin-7-methyl-ether (r = 0.82) and total phenolic content (r = 0.86) revealed a significant (*p* < 0.05) positive correlation with the DPPH assay. Apegenin-7-Glucoronide demonstrated a significant (*p* < 0.05) negative correlation (r = −0.81).

The use of Pearson’s correlation coefficients revealed significant correlations between several phenolic compounds and the antioxidant tests, proving the significance of these compounds and their contribution to the antioxidant power of the plant extract [11,38]. The interaction or synergistic effect among the polyphenolic compounds contained in thyme post-distilled residues may also contribute to their antioxidant capacity. It was the greater presence of these components that was responsible for the increased antioxidant capacity of the corresponding extracts. In conclusion, it can be confirmed that the concentrations of the chemical compounds of polyphenolic extracts have a signficant role in their antioxidative power.

## 3. Material and Methods

### 3.1. Plant Material

A total of 20 individual thyme shrubs were randomly collected from two wild populations (10 plants from Jbal kardmi Elkef (JKK) and 10 plants from Jbal Hantaya ElKef (JHK)) at the full bloom phenological stage. Voucher specimens of thyme were deposited at the Herbarium of Departamento de Desarrollo Rural, Enología y Agricultura Sostenible, Instituto Murciano de Investigación y Desarrollo Agrario y Medioambiental (IMIDA), La Alberca (Murcia, Spain). Fresh aerial parts of the individual plants were firstly dried at room temperature for ten days and afterwards dried in a forced-air drier at 35 °C for 48 h, until they reached a constant weight. 

### 3.2. Chemicals and Reagents

2,2-Diphenyl-1-picrylhydrazyl (DPPH•), Potassium persulfate, the Folin–Ciocalteu reagent, gallic acid and high purity standards were purchased from Sigma-Aldrich (Madrid, Spain). Methanol, acetonitrile, formic acid, anhydrous sodium carbonate and sodium acetate were supplied from Scharlau Chemie S.A. (Sentmenat, Spain). All reagents and solvents were purchased from Sigma-Aldrich (Madrid, Spain) with the exception of methanol (Honeywell, Germany). Authentic standards of gallic acid, salvianolic acid, caffeic acid, hesperidine, luteolin, rosmarinic acid, apigenin, genkwanin, quercetin and olivetol were obtained from Sigma-Aldrich (Madrid, Spain).

### 3.3. Preparation of the Plant Extracts

Individual plants were firstly distilled in a Clevenger system, after which, the oil-free distilled plant material was dried in a forced-air drier at 35 °C for 48 h (until it reached a constant weight) and then ground to pass through a 2 mm mesh. Dried samples (0.5 g) were extracted using 150 mL of methanol in a Soxhlet extractor (B-811) (Buchi, Flawil, Switzerland) for 2 h under a nitrogen atmosphere. Thyme extracts (TE) were taken to dryness at 35 °C under vacuum conditions in an evaporator system (SyncorePolyvap R-96) (Buchi, Flawil, Switzerland). The residue was redissolved in methanol and made up to 5 mL [11]. The yield of the extracts was expressed in terms of the milligrams of dry methanolic extract per gram of dry plant weight (mg DE/g DPW). Final extracts were kept in vials at −80 °C until their corresponding analyses were carried out (Figure 3).

### 3.4. Determination of the Total Polyphenolic Content

The total polyphenolic content (TPC) was determined by the Folin–Ciocalteu reagent method [39]. A reaction mixture of 15 μL of the thyme waste extract, 1185 μL of distilled water and 75 μL of the 10% Folin–Ciocalteu reagent were prepared. A vigorous stirring was performed and 225 μL of sodium carbonate (20%) was added. After 30 min of incubation, the absorbance of the resulting blue-colored solution was measured at 765 nm and 25 °C with a Shimadzu (UV-2401PC, Kyoto, Japan) spectrophotometer. A standard curve was produced by using different concentrations ranging from 0.1 to 1 mg/mL of gallic acid. The TPC was expressed as mg gallic acid equivalents per gram of dry plant weight (mg GAE/g DPW). Analyses were carried out in triplicate.

### 3.5. Antioxidant Activity

#### 3.5.1. DPPH•Radical-Scavenging Activity

The study of the DPPH• free radical scavenging activity of the thyme methanolic extracts was performed according to the method described by Brand-Williams et al. (1995) [40] with some modifications. Briefly, 50 μL and 100 μL of the sample was added to Eppendorf tubes containing 850 μL and 800 μL of methanol, respectively, and then 100 μL of 1 mM DPPH• was added. After 30 min of the reaction at 25 °C and protected from light, the scavenging activities of the samples and standards (Ascorbic acid, 1–100 mM in methanol) were evaluated by measuring the absorbance at 515 nm, in a Synergy MX UV–Vis spectrometer (BioTek Instruments Inc; Winooski, VT, USA).The absorbance of the control consisting of 900 μL of methanol and 100 μL of DPPH• solution was measured daily. Measurements were performed in triplicate. For each sample concentration tested, the inhibition percentage (%I) of DPPH• in the steady state was determined following the Equation:
**%I = [(Abs_control_ − Abs_sample_)/Abs_control_] × 100**

The results were expressed as the inhibitory concentration of the extract needed to decrease DPPH• absorbance by 50% (IC_50_). Concentrations are expressed in micrograms of dry plant extract per milliliter of methanol (IC_50_, μg/mL).

#### 3.5.2. Ferric-Reducing Antioxidant Power (FRAP)

The ferric-reducing ability of extracts was measured according to the method developed by Benzie and Strain (1996) [41]. Antioxidant compounds are able to reduce ferric iron, in the ferricyanide complex, Fe^3+^ to ferrous iron Fe^2+^, which develops a blue color. To prepare the FRAP reagent, a mixture of 0.1 mM acetate buffer (pH 3.6), 10 mM TPTZ in 40 mM HCl and 20 mM ferric chloride (10:1:1) was made. An aliquot of 40 μL of each sample was added to 1.2 mL of the FRAP reagent. The absorption was measured at 593 nm after 2 min of incubation at 37 °C. Measurements were performed in triplicate. Fresh working solutions of known Fe (II) concentrations (FeSO_4_.7H_2_O) of 0.05–1 mM were used to obtain the calibration curve and results were expressed as MFe^2+^ equivalent per milligram of dry plant weight (mMFe^2+^/ mg PDW).

### 3.6. Identification and Quantification of Polyphenolic Compounds by HPLC

Phenolic compounds were identified and quantified by HPLC. Chromatographic analyses were performed on a reverse phase ZORBAX SB-C18 column (4.6 × 250 mm, 5 μm pore size, Hewlett Packard, Palo Alto, CA, USA) using a guard column (ZORBAX SB-C184.6 × 125 mm, 5 μm pore size, Hewlett Packard, Palo Alto, CA, USA) at ambient temperature, based on the method adapted from Zheng and Wang [2]. Extracts were passed through a 0.45 μm filter (Millipore SAS, Molsheim, France) and 20 μL was injected in a Hewlett Packard (Germany) system equipped with a G1311A quaternary pump and G1315A photodiode array UV–Vis detector. The mobile phase was acetonitrile (A) and acidified water containing 0.05% formic acid (B). The gradient was as follows: 0 min, 5% A; 10 min, 15%A; 30 min, 25%A; 35 min, 30%A; 50 min, 55%A; 55 min, 90%A; 57 min, 100% A and then held for 10 min before returning to the initial conditions. The flow rate was 1.0 mL/min and the wavelengths of detection were set to 280 and 330 nm. The identification of the phenolic components was made through the comparison of retention times and spectra with those of commercially available standard compounds. For quantification, linear regression models were determined using standard dilution techniques. The results were expressed as mg of compound per gram of dry plant weight (mg/g DPW).

### 3.7. Statistical Analyses

All experiments were performed in triplicate (*n* = 3) and data were reported as means ± standard deviation (SD). A one-way ANOVA, followed by Duncan’s multiple range tests, was carried out to assess for significant differences between various experiments (a significant model was accepted for a *p*-value < 0.05) using Excel and STATISTICA software version 5.1. Pearson’s correlation coefficients were calculated. A *p* value less than 0.05 was considered to be statistically significant.

## 4. Conclusions

This study has investigated the total polyphenolic content, antioxidant activity and phenolic compounds of thyme post-distilled residues. These results proved that the plants with high levels of total polyphenolic content are characterized by high antioxidant capacity. The polyphenolic profile of thyme extracts revealed the presence of diosmin and rosmarinic acid as the most abundant compounds. The positive correlation between the polyphenolic content and the antioxidant capacity confirmed that the phenolic constituents are responsible for the antioxidant activity of thyme. The interaction or synergistic effect among the bioactive compounds contained in post-distilled thyme extract may also contribute to their antioxidant capacity. Thyme post-distilled residues have proven to be an effective potential source of polyphenols, as natural antioxidants beneficial properties to human health and could be useful in replacing or even decreasing synthetic antioxidants in foods, cosmetics and pharmaceutical products. This highlights the interest in extracting the phenolic compounds from the thyme by-products in order to exploit their biological activities (antioxidant, antimicrobial, anti-biofilm, anti-inflammatory and anticarcinogenic capacities). Therefore, supplementing a balanced diet with plant by-products may have beneficial health effects.

## Figures and Tables

**Figure 1 molecules-27-08791-f001:**
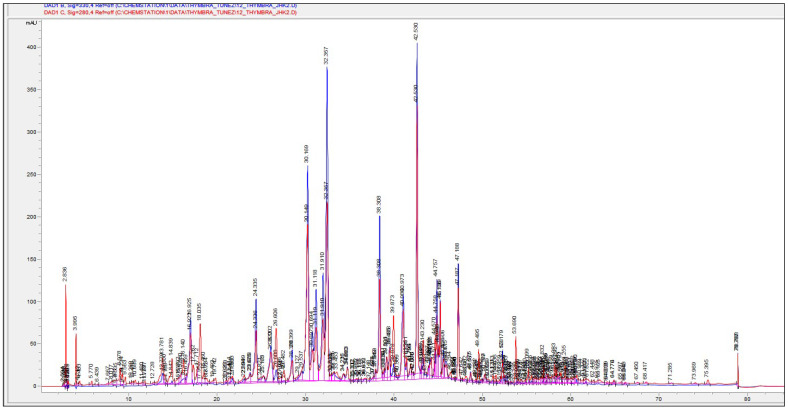
HPLC chromatogram of thyme residue extract at 280 nm and 330 nm with Retention Time.

**Figure 2 molecules-27-08791-f002:**
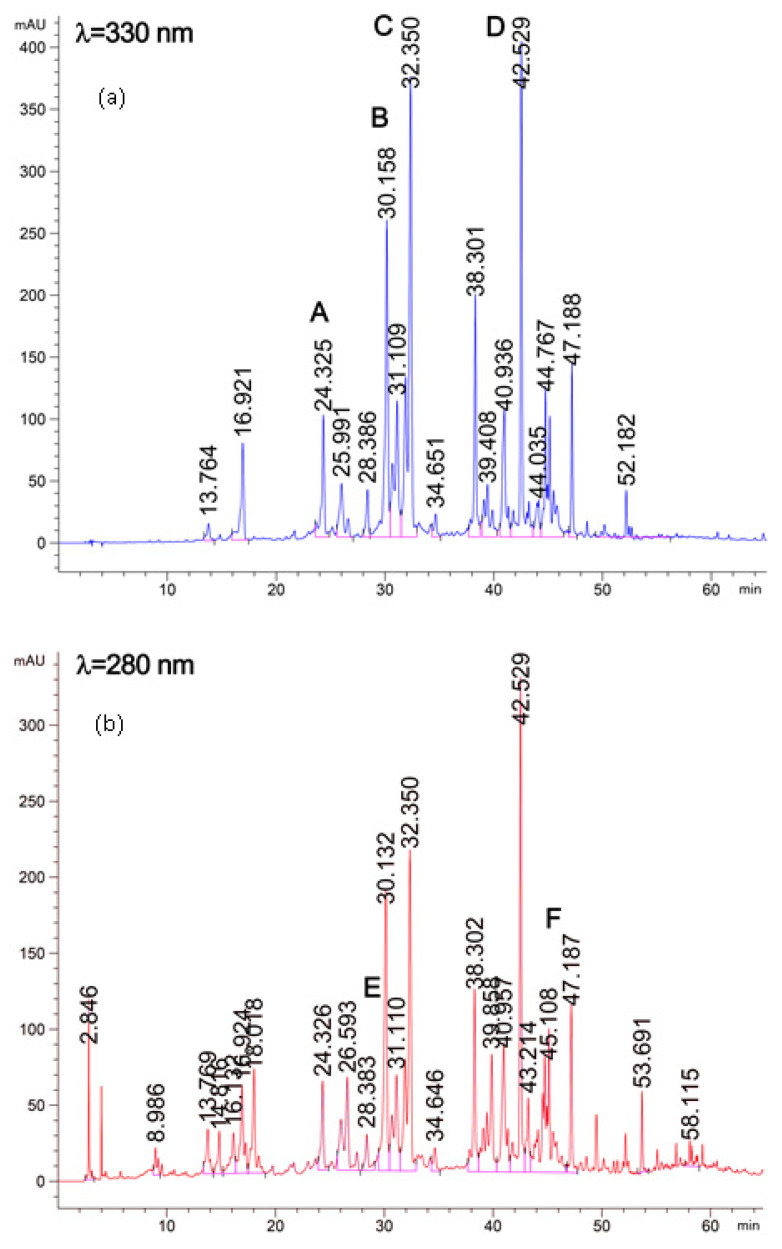
(**a**) Chromatogram and structure of the main identified compounds of thyme residues at **λ** = 330 nm: Luteolin-7-O-Neohesperidoside (A) Apigenin-7-Glucoronide (B), Rosmarinic acid (C), and Quercetin (D). (**b**) Chromatogram and structure of the main identified compounds of thyme residue extracts at **λ** = 280 nm: Diosmin (E), and Olivetol (F).

**Figure 3 molecules-27-08791-f003:**
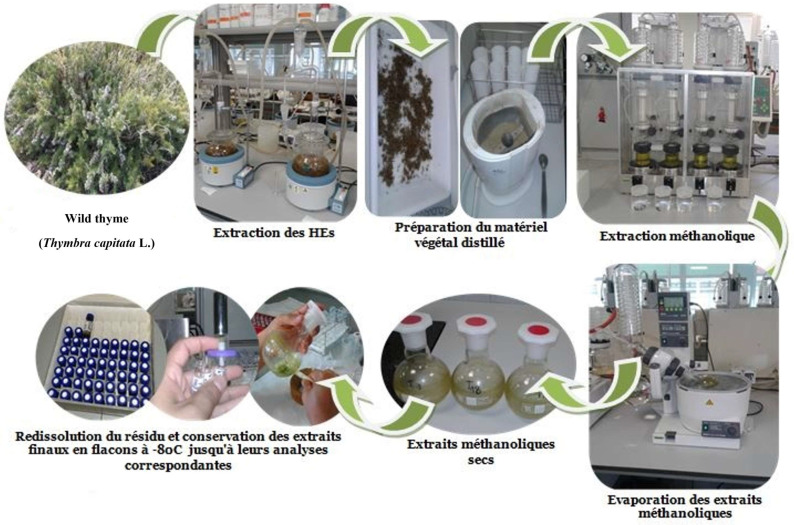
Preparation of methanolic extracts of post-distilled thyme.

**Table 1 molecules-27-08791-t001:** Extract yield, total polyphenolic content (TPC) and radical scavenging activity of thyme post-distilled residue extracts.

	Extract Yield (mg DME/g DPW)	Total Phenolic Content (TPC, mg GAE/g PDW)	DPPH (IC_50_, μg/mL)	FRAP (mMF^2+^/mg)
JKK	123.40 ± 13.27	126.7 ± 34.3	42.97 ± 2.10	50.21 ± 2.40
JHK	123.29 ± 16.55	107.84 ± 14.6	45.64 ± 2.29	42.22 ± 0.69

Code: Jbalkardmielkef (JKK), Jbalhantayaelkef (JHK). Results are expressed as means ± standard deviation (*n* = 10).

**Table 2 molecules-27-08791-t002:** HPLC polyphenolic profile of thyme post-distilled extracts.

Λ (nm)	RT (min)	Phenolic Compounds (mg/g)	Molecular Formula	JKK	JHK
280	10	Salvianolic acid	C_26_H_22_O_10_	0.93 ± 0.10 ^a^	0.93 ± 0.24 ^a^
330	18.81	Caffeic acid	C_9_H_8_O_4_	0.92 ± 0.10 ^a^	0.61 ± 007 ^b^
330	26.66	Luteolin-7-O-Neohesperidoside	C_27_H_30_O_15_	7.10 ± 1.61 ^a^	7.25 ± 1.24 ^a^
330	30.77	Apigenin-7-Neohesperidoside	C_27_H_30_O_14_	2.75 ± 0.76 ^a^	3.13 ±0.90 ^a^
280	32.54	Diosmin	C_28_H_32_O_15_	33.80 ± 15.90 ^a^	24.26 ± 6.65 ^a^
330	33.3	Apegenin-7-Glucoronide	C_21_H_18_O_11_	6.23 ± 1.31 ^a^	7.94 ± 2.09 ^b^
330	34.62	Rosmarinic acid	C_18_H_16_O_8_	26.29 ± 3.66 ^a^	22.01 ± 3.23 ^b^
330	39.81	6-hydroxy-apegenin	C_15_H_10_O_6_	4.22 ± 2.08 ^a^	3.97 ± 1.59 ^a^
280	41.26	Eridictyol	C_15_H_12_O_6_	2.62 ± 0.45 ^a^	2.71 ± 0.42 ^a^
330	42.17	Quercetin	C_15_H_10_O_7_	6.61 ± 0.74 ^a^	5.81 ± 1.38 ^a^
330	45.52	Apigenin	C_15_H_10_O_5_	1.44 ± 0.31 ^a^	1.43 ± 0.19 ^a^
330	49.13	Luteolin-7-methyl-ether	C_16_H_12_O_6_	0.76 ± 0.08 ^a^	0.59 ± 0.16 ^b^
280	50.06	Olivetol	C_11_H_16_O_2_	5.76 ± 1.02 ^a^	5.26 ±1.00 ^a^
330	52.55	Genkwanin	C_16_H_12_O_5_	0.89 ± 0.20 ^a^	069 ± 0.15 ^b^

Code: Jbal kardmi Elkef (JKK), Jbal Hantaya ElKef (JHK). RT: Retention time. Contents of phenolic compounds expressed as mg of compound /g of dry plant weight (mg of compound/g DPW). Results are expressed as means ± standard deviation (*n* = 10). The different lowercase letters (a,b) in the same row indicate significantly different values (*p* < 0.05).

**Table 3 molecules-27-08791-t003:** Linear correlation coefficients established between phenolics and the DPPH antioxidant activity.

Phenolic Compounds	DPPH
Salvianolic acid	−0.13
Caffeic acid	0.93 *
Luteolin-7-O-Neohesperidoside	−0.10
Apigenin-7-Neohesperidoside	−0.23
Diosmin	0.60
Apegenin-7-Glucoronide	−0.81 *
Rosmarinic acid	0.83 *
6-hydroxy-apegenin	0.48
Eridictyol	−0.31
Quercetin	0.78
Apigenin	0.30
Luteolin-7-methyl-ether	0.82 *
Olivetol	0.24
Genkwanin	0.50
TPC	0.86 *

* Significant correlation at *p* < 0.05.

## Data Availability

Not applicable.

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
