# Peer review of "Polyphenolic Profile of Tunisian Thyme (Thymbra capitata L.) Post-Distilled Residues: Evaluation of Total Phenolic Content and Phenolic Compounds and Their Contribution to Antioxidant Activity"

_molecules, 2022, doi:10.3390/molecules27248791_

Round 1
Reviewer 1 Report
Line 58-60: Do BHT and BHA promote carcinogenesis or it has anti-carcinogenic effect? Make clear? Split the sentences.
Line 65-66: “antioxidant, anti-inflammatory, anti-microbial and anti-carcinogenic activities” include references
Also, include a section on the antioxidant properties of plant products in general in the introduction. The authors may refer to the following publications (https://doi.org/10.1007/s00203-021-02650-7, https://doi.org/10.2174/1574892816666210401143750 , https://doi.org/10.1038/s41598-021-89437-4, https://doi.org/10.1155/2021/9917810)
Line 94-96: “For this reason, it is important to develop methods that allow the extraction of all phenolic compounds. It seems that the increase in temperature, in the range of 50-70 °C, produces a higher solubility of the compounds and increases their rate of transfer ” What is the significance of this sentence as the study doesn’t include the method development for the extraction of bound phenolics?
Antioxidant Assay:
How can the authors explain the strong antioxidant property with a single assay like DPPH scavenging?
How the IC 50 values have been calculated with just two concentrations?
The DPPH assay used two different concentrations in µl, but the IC 50 values are in µg/ml. How is this possible?
The Graphs and figures should b included
HPLC: What are the limit of detection (LOD) and limit of quantification (LOQ) for the compounds? HPLC chromatograms are missing.
The actual p-value of HPLC data should be indicated in the table.
Why haven’t the authors estimated the flavonoid concentration?
References: Need to add more recent references
Author Response
Response to reviewers and editor comments
Dear Editor and Reviewers
In my name, Dr. Kheiria Hcini, and in the names of all the authors, we would like to thank you very much for the time you have devoted and the effort you have provided to correct this manuscript.
Firstly, we would like to thank you for your kind letter and for reviewers’ constructive comments concerning our article (Manuscript ID: molecules-2058195, Polyphenolic profile of Tunisian thyme (Thymbra capitata L.) post-distilled residues: Evaluation of total phenolic content and phenolic compounds and their contribution to antioxidant activity). These comments are all valuable and helpful for improving our article. All the authors have seriously discussed about all these comments. According to the reviewers’ comments, we have tried best to modify our manuscript to meet with the requirements of your journal. All revisions of our manuscript were marked using “Track changes” function and highlighted by using yellow colored text. Also the revision of English language was highlighted by using pink colored text. Point-by-point responses to the reviewers are listed below this letter.
Secondly, if there are any other modifications we could make, we would like very much to modify them and we really appreciate your help. Thank you very much for your help.
Responses to Comments and Suggestions of Reviewer 1
Line 58-60: Do BHT and BHA promote carcinogenesis or it has anti-carcinogenic effect? Make clear? Split the sentences.
* BHT and BHA promote carcinogenesis and they do not have an anti-carcinogenic effect. Sentences have been corrected.
Line 65-66: “antioxidant, anti-inflammatory, anti-microbial and anti-carcinogenic activities” include references.
Also, include a section on the antioxidant properties of plant products in general in the introduction. The authors may refer to the following publications (https://doi.org/10.1007/s00203-021-026507, https://doi.org/10.2174/1574892816666210401143750 , https://doi.org/10.1038/s41598-021-89437-4, https://doi.org/10.1155/2021/9917810).
* Some new references have been included
Line 94-96: “For this reason, it is important to develop methods that allow the extraction of all phenolic compounds. It seems that the increase in temperature, in the range of 50-70 °C, produces a higher solubility of the compounds and increases their rate of transfer ” What is the significance of this sentence as the study doesn’t include the method development for the extraction of bound phenolics?
* In certain cases, cell wall phenolics or bound phenolics could be released consequently to heat exposure, thus generating more phenolics to be extracted.
Antioxidant Assay:
How can the authors explain the strong antioxidant property with a single assay like DPPH scavenging?
* Aromatic and medicinal plants, such as species belonging to the family Lamiaceae, are known for their antioxidant power, evaluated by various tests (DPPH, FRAP, ABTS, etc.).
the FRAP test has been added (I did this test in parallel with DPPH test and thought that the better known and more used DPPH test is sufficient to present it in the results section).
How the IC 50 values have been calculated with just two concentrations?
** 1. Dried post-stilled samples (thym1-20), (0.5 g) were extracted using 150 mL of methanol.
- Thyme extracts (TE) were taken to dryness at 35 °C under vacuum conditions in an evaporator system (Syncore Polyvap R-96) (Buchi, Flawil, Switzerland). The residue was re-dissolved in methanol and made up to 5 mL.
- in the first time, we performed a DPPH test with 5 volumes (20, 40, 60, 80, and 100 μL) of thyme extract (diluted 20 times) and we found that the volume 50 μL has an inhibition of DPPH less than 50% and 100 μL has an inhibition greater than 50%. So we have chosen these two volumes (two concentrations). For each sample we have done three replicates.
The DPPH assay used two different concentrations in µl, but the IC 50 values are in µg/ml. How is this possible?
- And here is the explanation of the calculation
- Reaction mixture: μL 50 μL thyme extract (TE) + 850 μL methanol + 100 μL DPPH, incubation 30 mn at the darck, 25 °C.
- DO du sample (TE + DPPH) et DO of control (only DPPH + methanol)
- I%: (DO Control – DO sample) / DO control
- Curve: I% en function of Volume (thyme extract)
- Curve equation: y = ax (y = I%, x = volume of the extract)
- From the equation of the curve we must calculate the necessary volume of the TE to make a 50% inhibition of the activity of DPPH, so x = y/a = 50/a
-For thym1 x = 50/0.619 = 80.775 μL (dilution 20), so the necessary volume of TE = 80.775/20 = 4.038 μL
-[0.0523g = 52.3mg dans 5mL], the concentration initial of thym1
52.3 mg TE ...... 5000 μL
X mg TE……..4 μL
X = (52.3 mg x 4 μL)/5000 μL = 0.0422 mg = 42.2 μg
So IC 50 DPPH = 42.2 μg /ml (volume of reaction = 1 mL)
The Graphs and figures should b included
* Please can explain to me this comment
HPLC: What are the limit of detection (LOD) and limit of quantification (LOQ) for the compounds? HPLC chromatograms are missing.
* HPLC chromatograms have been included (figure 1, figure 2 a and b).
The actual p-value of HPLC data should be indicated in the table.
* p-value of HPLC is indicated below the table
Why haven’t the authors estimated the flavonoid concentration?
* The objective of this present work is to show and prove the richness of post-distilled thyme residues in total polyphenolic content, which are known for their potential antioxidant power, and the identification of phenolic compounds. So we only have estimated the TPC.
References: Need to add more recent references
* Some recent references were added
Reviewer 2 Report
1. Every important concluding statement in the introduction section needs to be supported by literature.
2. Although there are many polyphenols in the natural products described, the authors are suggested to select the components considered more important and characterize their structure.
3. Therefore, the authors are encouraged to describe about the synergistic or antagonistic effect of those various important components in antioxidant effect. For example, please choose two or three components that are considered more important to discuss or cite literature.
4. The experimental conditions need to be specified in detail, the concentration, the reaction temperature is required.
Author Response
Response to reviewers and editor comments
Dear Editor and Reviewers
In my name, Dr. Kheiria Hcini, and in the names of all the authors, we would like to thank you very much for the time you have devoted and the effort you have provided to correct this manuscript.
Firstly, we would like to thank you for your kind letter and for reviewers’ constructive comments concerning our article (Manuscript ID: molecules-2058195, Polyphenolic profile of Tunisian thyme (Thymbra capitata L.) post-distilled residues: Evaluation of total phenolic content and phenolic compounds and their contribution to antioxidant activity). These comments are all valuable and helpful for improving our article. All the authors have seriously discussed about all these comments. According to the reviewers’ comments, we have tried best to modify our manuscript to meet with the requirements of your journal. All revisions of our manuscript were marked using “Track changes” function and highlighted by using yellow colored text. Also the revision of English language was highlighted by using pink colored text. Point-by-point responses to the reviewers are listed below this letter.
Secondly, if there are any other modifications we could make, we would like very much to modify them and we really appreciate your help. Thank you very much for your help.
Response to comments and suggestions for Reviewer 2
- Every important concluding statement in the introduction section needs to be supported by literature.
* The literature of introduction have been revised
- Although there are many polyphenols in the natural products described, the authors are suggested to select the components considered more important and characterize their structure.
*Thank you so much for your suggestion, it is a good idea. We have selected the more important considered components and we have characterized their structure (please find the structure and HPLC profile chromatogram included in the text).
- Therefore, the authors are encouraged to describe about the synergistic or antagonistic effect of those various important components in antioxidant effect. For example, please choose two or three components that are considered more important to discuss or cite literature.
* Two components rosmarinic acid and diosmin, that are considered more important, have been choose to be discuss.
- The experimental conditions need to be specified in detail, the concentration, the reaction temperature is required.
* The experimental conditions, the concentration, and the reaction temperature have been highlighted by using yellow colored text.
Round 2
Reviewer 1 Report
No more comments
Reviewer 2 Report
The manuscript has been improved. I have no further questions.